# Inequalities in healthcare disruptions during the COVID-19 pandemic: evidence from 12 UK population-based longitudinal studies

Jane Maddock [1], Sam Parsons [2], Giorgio Di Gessa [3], Michael J Green [4], Ellen J Thompson [5], Anna J Stevenson [6], Alex SF Kwong [7,8], Eoin McElroy,[9] Gillian Santorelli,[10] Richard J Silverwood [2], Gabriella Captur [1], Nishi Chaturvedi [1], Claire J Steves [5], Andrew Steptoe [3], Praveetha Patalay [1,2], George B Ploubidis [2], Srinivasa Vittal Katikireddi [4]

JM, SP, GDG, MJG and EJT contributed equally.

For numbered affiliations see end of article.

**Correspondence to**
Dr Jane Maddock;
jane.maddock@ucl.ac.uk

## ABSTRACT

**Objectives** We investigated associations between multiple sociodemographic characteristics (sex, age, occupational social class, education and ethnicity) and self-reported healthcare disruptions during the early stages of the COVID-19 pandemic.

**Design** Coordinated analysis of prospective population surveys.

**Setting** Community-dwelling participants in the UK between April 2020 and January 2021.

**Participants** Over 68 000 participants from 12 longitudinal studies.

**Outcomes** Self-reported healthcare disruption to medication access, procedures and appointments.

**Results** Prevalence of healthcare disruption varied substantially across studies: between 6% and 32% reported any disruption, with 1%–10% experiencing disruptions in medication, 1%–17% experiencing disruption in procedures and 4%–28% experiencing disruption in clinical appointments. Females (OR 1.27; 95% CI 1.15 to 1.40; I²=54%), older persons (eg, OR 1.39; 95% CI 1.13 to 1.72; I²=77% for 65–75 years vs 45–54 years) and ethnic minorities (excluding white minorities) (OR 1.19; 95% CI 1.05 to 1.35; I²=0% vs white) were more likely to report healthcare disruptions. Those in a more disadvantaged social class were also more likely to report healthcare disruptions (eg, OR 1.17; 95% CI 1.08 to 1.27; I²=0% for manual/routine vs managerial/professional), but no clear differences were observed by education. We did not find evidence that these associations differed by shielding status.

**Conclusions** Healthcare disruptions during the COVID-19 pandemic could contribute to the maintenance or widening of existing health inequalities.

## INTRODUCTION

The COVID-19 pandemic has affected all aspects of society. Health systems worldwide have faced major disruption as they respond to large increases in demand arising from the

---

### STRENGTHS AND LIMITATIONS OF THIS STUDY

⇒ We conducted coordinated primary analyses in 12 UK longitudinal population studies, and pooled results using a random effects meta-analysis.

⇒ Use of multiple studies increased statistical power to look at subpopulations such as ethnic minority groups across cohorts and allowed for greater examination of how inequalities were patterned by age.

⇒ Most studies were weighted to be representative of their target ages in the UK population, and findings were robust to excluding those that were not.

⇒ We did not adjust for whether respondents needed healthcare, so the inequalities observed may be at least partly attributable to inequalities in needing healthcare.

⇒ Data on prepandemic healthcare disruption were not available, so we could not tell if inequalities in healthcare disruption had widened or narrowed during the pandemic.

---

COVID-19 disease.[1–5] Furthermore, healthcare access has been reduced by governmental control measures and the public's fear of contracting infection.[6] Disruptions may have both short-term and long-term health consequences as preventive treatments are foregone, disease surveillance is interrupted and disease diagnoses are delayed. While the disruption of health systems can impact the entire population, it has become apparent that not all groups have been affected equally. For example, recent evidence has demonstrated that both elective and emergency hospital admissions vary by socioeconomic deprivation and ethnic minority quintiles, with the more deprived areas showing a large fall in elective admissions, and areas with

high ethnic minority populations showing larger falls in emergency admissions.[5] Understanding the impacts of the pandemic on health systems and on equity of healthcare access is therefore a major policy priority.

In the UK, the National Health Service (NHS) provides free healthcare and prioritises equity of delivery. However, the UK's relatively high COVID-19 burden and associated repeated lockdown measures have raised concerns that the health system may not be providing accessible care to those who need it most. Recent reports from NHS Digital indicate a large increase in those waiting 12 months or more for elective treatments in February 2021 compared with March 2020.[7] Furthermore, despite decreases in attendance at accident and emergency (A&E) services,[4] the number of patients waiting over 12 hours for admission was 34% higher in January 2021 than January 2020. Disruption to pharmacological treatments has also been reported with delays in accessing medication.[8 9] However, a comprehensive assessment of inequalities in healthcare disruption in the community is lacking.

It is well known that health systems do not meet the needs of all social groups equitably, with marked health inequalities by sex, ethnicity and socioeconomic position.[10 11] For example, the inverse care law demonstrates that health service provision is often not allocated according to need, with more socioeconomically deprived areas relatively underserved.[12] Given the barriers that some social groups face in accessing high-quality healthcare, there is considerable concern that disadvantaged groups (eg, ethnic minorities) will be disproportionately impacted by healthcare disruption during the COVID-19 pandemic, as some emerging evidence suggests.[13 14]

Harnessing multiple longitudinal studies allows inequalities to be studied in detail by improving statistical power and allows consistency of findings to be investigated. We therefore aimed to investigate inequalities in healthcare disruption during the COVID-19 pandemic in 12 population-based longitudinal studies, to help inform targeting of policy responses as we move out of the acute phases of the pandemic. We investigate healthcare disruptions (including prescription or medication access, procedures or surgery, clinical appointments) by sex, age, ethnicity, education and occupational social class and we explore whether associations differ by age, or for those who have been recommended to 'shield' due to clinical vulnerability.

## METHODS
### Design
The UK National Core Studies–Longitudinal Health and Wellbeing programme aims to draw together data from multiple UK population-based longitudinal studies to answer questions relevant to the pandemic response. By coordinating analyses within each study and statistically pooling results in a meta-analysis, we can provide robust evidence to understand healthcare disruptions during the pandemic.

### Participants
Data were from 12 UK population studies which had conducted surveys both before and during the COVID-19 pandemic. Details of the design, sample frames, current age range, timing of the COVID-19 surveys, response rates and analytical sample size are available in online supplemental table S1 in supplementary file 4.

Our population of interest is the current UK population aged 16 years or older. The following studies are considered to be nationally representative samples of their target age groups: the Millennium Cohort Study (MCS)[15]; Next Steps (NS)[16]; the 1970 British Cohort Study (BCS70)[17]; the National Child Development Study (NCDS)[18]; the National Survey of Health and Development (NSHD)[19 20]; Understanding Society (USOC)[21]; and the English Longitudinal Study of Ageing (ELSA).[22] We also included the Avon Longitudinal Study of Parents and Children (ALSPAC-G1)[23]; the parents of the ALSPAC-G1 cohort which we refer to as ALSPAC-G0[24]; the Born in Bradford (BIB) study[25 26]; Generation Scotland: the Scottish Family Health Study (GS)[27]; and the UK Adult Twin Registry (TwinsUK).[28 29] We present the results from all 12 studies in the main manuscript and results restricted to representative samples in online supplemental file 1.

We can further categorise these studies into age-homogenous birth cohorts (where all individuals were of similar age within each cohort) and age-heterogeneous studies (each covering a range of age groups). The age-homogenous studies include MCS, ALSPAC-G1, NS, BCS, NCDS and NSHD. The age-heterogenous studies include BIB, USOC, GS, ALSPAC-G0, TwinsUK and ELSA. Analytical samples were defined within each study based on respondents who had no missing data on at least one healthcare disruption outcome in a COVID-19 survey and on a minimum set of covariates (sex, ethnicity and age where relevant). Most studies were weighted to be representative of their target populations accounting for differential non-response.[20 30 31] Weights were not available for BIB or TwinsUK. Studies were ordered for presentation by age of sample (youngest to oldest), with the age-homogenous cohorts first, followed by the age-heterogenous studies. Missing data within surveys were generally low, especially for healthcare disruption variables, but approximately 5%–10% of respondents across studies were excluded due to missing baseline covariates.

### Measures
Below we describe the overall approach to measuring each variable in the analysis.

### Outcomes
We assessed self-reported disruptions to prescriptions or medication access; procedures or surgery; and appointments (eg, with a general practitioner or outpatient services); and a combined variable indicating disruptions to any of the aforementioned. Any deviation from planned or existing treatment was coded as a disruption, regardless of the reason for the disruption. The wording

of the questions was the same for MCS, NS, BCS70, NCDS and NSHD. There was variation in how the questions were asked in the other studies. Full details of the questions and coding used within each study are available in online supplemental file 2. ALSPAC did not have information about prescriptions or medication access. BIB did not have information about procedures or surgery. TwinsUK did not have information about procedures or surgery or appointments. Where multiple pandemic survey waves had been included, we coded for any disruptions reported up to and including the most recent. This meant at least 7 months of follow-up for most studies (GS had five and ELSA had four, while ALSPAC had the longest follow-up period at 9 months). Online supplemental table S3 shows how the prevalence for any experience of each disruption accumulated across the six USOC surveys. The majority of those who experienced each type of healthcare disruption had already experienced it by the end of May 2020.

## Indicators of inequality

We assessed inequalities associated with key sociodemographic characteristics, that is, sex, age, ethnicity, education and occupational social class. For age, we considered age groups categorised as: 16–24; 25–34; 35–44; 45–54; 55–64; 65–74; and 75+ years. Depending on the level of detail of ethnicity available, we examined both a binary (white (including white minorities) vs ethnic minorities (excluding white minorities)) and a finer categorisation of ethnicity (white, south asian, black, mixed, other asian, other ethnic minority). For education, we distinguished between degree or equivalent; A-level or equivalent (ie, post-compulsory schooling qualifications); General Certificate of Secondary Education (GCSE) or equivalent (ie, qualifications for completing compulsory schooling); and fewer or no qualifications. We also examined occupational class with the following categories (based on different coding schemes in different studies): professional/managerial; intermediate; routine/manual; and other (which included never/long-term non-employed and, in some studies, respondents who could not be classified elsewhere). Respondents' education and occupational class were not available in the MCS or ALSPAC-G1, so we considered parental education or household social class. For full details, see online supplemental file 2.

## Moderators

We decided *a priori* to examine modification by age and clinical vulnerability to COVID-19 to see whether inequalities varied by life stage or were particularly acute for those with higher healthcare needs and at higher risk from COVID-19 harms. For moderation by age, the age-heterogeneous studies split their samples into the age bands covered, while age-homogeneous cohorts were included within the appropriate age bands (see above for banding). In the UK, clinically extremely vulnerable people were advised to stay at home ('shield') during the pandemic. Respondents were directly asked whether they had received a letter from the NHS advising them to stay at home and protect themselves. Specific survey questions can be found in online supplemental file 2.

## Other variables

The following covariates were also included where relevant and available within each study: UK nation (ie, England, Scotland, Wales or Northern Ireland); household composition (based on partnership status and whether there were children in the household); and prepandemic self-reported health (good vs poor).

## Analysis

Within each study, distributions of sociodemographic characteristics and healthcare disruption were examined. Then, each healthcare disruption outcome was regressed on each indicator of inequality (ie, sex, age, ethnicity, education and occupational class). Unadjusted associations are included in online supplemental file 3. Since our aim was primarily to describe inequalities, we focus on presenting associations with minimal adjustment only for sex, age and ethnicity when applicable. To assess whether associations were independent of other related factors, we also provide results in online supplemental file 3 for any healthcare disruption which additionally adjust for education, occupational class, UK nation (where appropriate), household composition and prepandemic self-reported health. Moderation by age and shielding status was assessed using stratified models.

Results were then meta-analysed for each outcome for the full sample, and within age and shielding strata. We used a random effects meta-analysis with restricted maximum likelihood. For stratified results, a test of group differences was performed using the subgroup meta-analysis command. We report heterogeneity using the $I^2$ statistic (0% indicates low variation between estimates across studies, while values closer to 100% indicate greater heterogeneity).

Finally, in sensitivity analyses we restricted the meta-analyses to representative studies (MCS, NS, BCS70, NCDS and NSHD, USOC and ELSA). Meta-analyses were conducted in Stata V.16.[32]

## Patient and public involvement

None.

## RESULTS
### Descriptive statistics

The distribution of demographic and socioeconomic characteristics within each study is presented in table 1. A total of 68 912 participants were included in the coordinated analysis. Due to study design, participants from BIB were all female, as were the vast majority (89.4%) from TwinsUK. The age ranged from 16 years in BIB and USOC to 90+ years in TwinsUK and ELSA.

Overall, the prevalence of any healthcare disruption ranged from 6.4% in TwinsUK to 31.8% in USOC (figure 1). Table 2 shows that disruptions to medical

**Table 1** Per cent (and n) distribution of demographic and socioeconomic characteristics by study

| | MCS | ALSPAC-G1 | NS | BCS70 | NCDS | NSHD | BIB | USOC | GS | ALSPAC-G0 | TwinsUK | ELSA |
|---|---|---|---|---|---|---|---|---|---|---|---|---|
| Total analytic, n | 3147 | 3430 | 3311 | 5175 | 5747 | 1569 | 1726 | 13 253 | 17 139 | 3625 | 4282 | 6508 |
| Female | 65.0 (2045) | 65.3 (2240) | 64.8 (2145) | 57.9 (2994) | 53.7 (3086) | 52.6 (825) | 100.0 (1726) | 57.9 (7668) | 67.0 (11 476) | 73.1 (2651) | 89.4 (3830) | 56.3 (3663) |
| Mean age in 2020 (range) | 19.5 (18.7–20.1) | 28.4 (27–29) | 30.6 (29.9–31.4) | 50.5 (50.4–50.6) | 62.6 (62.5–62.7) | 74 | 37.5 (16–54) | 51.1 (16–96.2) | 57.0 (18–100) | 59.4 (45–89) | 61.2 (22–96) | 69.3 (52–90+) |
| **Ethnicity** | | | | | | | | | | | | |
| White | 86.1 (2708) | 98.4 (3330) | 74.6 (2470) | NA | NA | NA | 37.8 (653) | 98.3 (16 843) | 87.2 (11 561) | 98.4 (3567) | 97.1 (4156) | 95.9 (6239) |
| South Asian | 7.6 (240) | NA | 15.0 (496) | NA | NA | NA | 56.1 (968) | 0.4 (70) | 6.7 (885) | NA | 0.7 (28) | 2.1 (135) |
| East Asian | 1.0 (30) | NA | NA | NA | NA | NA | NA | 0.3 (51) | 1.2 (155) | NA | 0.1 (3) | NA |
| Black | 2.6 (83) | NA | 3.8 (127) | NA | NA | NA | 2.0 (34) | 0.1 (21) | 2.5 (334) | NA | 1.1 (45) | 1.2 (75) |
| Mixed | 2.4 (76) | NA | 4.6 (152) | NA | NA | NA | 1.4 (24) | 0.6 (105) | 1.8 (241) | NA | 0.9 (38) | 0.9 (59) |
| Other | 0.3 (10) | NA | 2.0 (66) | NA | NA | NA | 2.7 (47) | 0.3 (49) | 0.6 (77) | NA | 0.3 (12) | NA |
| All ethnic minorities | 13.9 (439) | 2.9 (100) | 25.4 (841) | NA | NA | NA | 62.2 (1073) | 1.3 (226) | 12.8 (1692) | 1.6 (58) | 2.9 (126) | 4.1 (269) |
| **Education** | | | | | | | | | | | | |
| Higher education or degree | 55.9 (1758) | 29.0 (994) | 48.9 (1620) | 46.6 (2411) | 46.0 (2646) | 29.0 (994) | 35.1 (556) | 50.7 (8602) | 47.1 (6238) | 29.7 (1075) | 55.7 (2386) | 25.6 (1666) |
| A-level or equivalent | 15.0 (473) | 35.1 (1203) | 23.4 (773) | 14.2 (733) | 18.0 (1034) | 35.1 (1203) | 17.2 (273) | 35.9 (6096) | 11.6 (1543) | 29.7 (1078) | 11.6 (498) | 27.6 (1798) |
| GCSE or equivalent | 19.5 (615) | 26.1 (896) | 19.0 (628) | 23.4 (1209) | 22.8 (1311) | 26.1 (896) | 22.3 (354) | 6.2 (1046) | 25.2 (3341) | 30.3 (1098) | 20.5 (877) | 22.3 (1452) |
| <GCSE or none | 9.6 (301) | 9.83 (337) | 8.8 (290) | 15.9 (822) | 13.2 (756) | 9.8 (337) | 25.5 (405) | 7.2 (1214) | 16.1 (2131) | 10.3 (374) | 12.2 (521) | 24.5 (1592) |
| **Social class** | | | | | | | | | | | | |
| Managerial, admin, professional | 51.3 (1614) | 18.0 (616) | 47.6 (1575) | 42.7 (2209) | 23.0 (1319) | 18 (616) | 31.2 (475) | 81.0 (10 716) | 35.0 (4639) | 13.4 (486) | NA | 32.4 (2111) |
| Intermediate | 15.4 (484) | 46.2 (1583) | 18.9 (625) | 21.1 (1091) | 14.9 (856) | 46.1 (1583) | 35.7 (545) | 14.4 (1906) | 17.1 (2264) | 41.2 (1492) | NA | 23.0 (1497) |
| Manual/routine | 18.9 (595) | 35.3 (1212) | 15.0 (495) | 19.5 (1009) | 16.5 (948) | 35.3 (1212) | 25.3 (386) | 4.4 (581) | 20.1 (2663) | 44.6 (1617) | NA | 28.2 (1834) |
| Other | 14.4 (454) | 0.6 (19) | 18.6 (616) | 16.7 (866) | 45.7 (2624) | 0.6 (19) | 7.8 (119) | 0.2 (27) | 27.8 (3687) | 0.8 (30) | NA | 16.4 (1066) |
| Instructed to shield | 2.5 (79) | NA | 3.3 (110) | 5.2 (267) | 6.9 (393) | 8.8 (101) | 7.6 (131) | 6.2 (825) | 7.8 (1332) | NA | 5.9 (252) | 16.3 (1062) |

Sources: Millennium Cohort Study (MCS); Children of the Avon Longitudinal Study of Parents and Children (ALSPAC-G1); Next Steps (NS); 1970 British Cohort Study (BCS70); National Child Development Study (NCDS); National Survey of Health and Development (NSHD); Born in Bradford (BIB); Understanding Society (USOC); Generation Scotland: the Scottish Family Health Study (GS); parents of ALSPAC (ALSPAC-G0); UK Adult Twin Registry (TwinsUK); English Longitudinal Study of Ageing (ELSA).

Studies are ordered by age homogeneity/heterogeneity and mean age of respondents at the time of the interview. Samples for each study are restricted to respondents with non-missing information on healthcare disruptions and valid information on sex, social class, education and (where applicable) age and ethnicity. All information about how data were collected and variables were coded is available in online supplemental file 2.

Unweighted data.

GCSE, General Certificate of Secondary Education; NA, not available/info not collected.

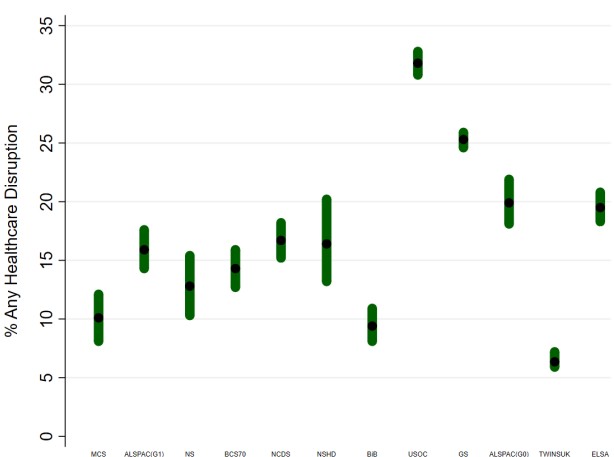

**Figure 1** Prevalence (and 95% CIs) of any healthcare disruption by study. Sources: Millennium Cohort Study (MCS); Children of the Avon Longitudinal Study of Parents and Children (ALSPAC-G1); Next Steps (NS); 1970 British Cohort Study (BCS70); National Child Development Study (NCDS); National Survey of Health and Development (NSHD); Born in Bradford (BIB); Understanding Society (USOC); Generation Scotland: the Scottish Family Health Study (GS); parents of ALSPAC (ALSPAC-G0); UK Adult Twin Registry (TwinsUK); English Longitudinal Study of Ageing (ELSA). Studies are ordered by age homogeneity/heterogeneity and mean age of respondents at the time of the interview. Samples for each study were restricted to respondents with non-missing information on healthcare disruptions and valid information on sex, social class, education and (where applicable) age and ethnicity. All information about how data were collected and variables were coded is available in online supplemental file 2.

appointments were most common, ranging from 3.5% (ELSA) to 28.4% (USOC). Disruptions in prescriptions or medication access varied from 0.8% (ELSA) to 10.4% (GS). Disruptions to procedures or surgery were least common ranging from 0.7% (MCS) to 16.8% (ELSA).

The following sections describe the results adjusted for sex, age and ethnicity when applicable. Unadjusted results and results adjusted for education, occupational class, UK nation (where appropriate), household composition and prepandemic self-reported health can be found in online supplemental file 3. The associations were largely robust to further adjustment.

### Sex and healthcare disruptions

Across all studies, females were generally more likely to report any healthcare disruptions than males (see online supplemental table S4 for details).

Pooled results from the meta-analysis demonstrate that females had increased odds of any healthcare disruption compared with males (OR 1.27; 95% CI 1.15 to 1.40; $I^2=54\%$, figure 2, online supplemental file 3). Similar associations were observed for disruptions to appointments (OR 1.33; 95% CI 1.17 to 1.52; $I^2=60\%$). The association between sex and disruptions to procedures and

**Table 2** Per cent prevalence (and 95% CIs) of healthcare disruptions during the pandemic by study

| | MCS | ALSPAC-GI | NS | BCS 70 | NCDS | NSHD | BIB | USOC | GS | ALSPAC-G0 | TwinsUK | ELSA |
|---|---|---|---|---|---|---|---|---|---|---|---|---|
| Prescription/ medication access | 4.0 (2.3 to 5.5) | NA | 3.8 (2.3 to 5.3) | 3.4 (2.7 to 4.2) | 2.4 (1.8 to 3.0) | 2.2 (1.3 to 3.8) | 1.2 (0.7 to 1.7) | 5.5 (5.0 to 6.1) | 10.4 (9.9 to 10.9) | NA | 2.9 (2.5 to 3.3) | 0.8 (0.6 to 1.2) |
| Procedures or surgery | 0.7 (0.0 to 1.2) | 1.6 (1.2 to 2.1) | 2.1 (0.0 to 3.8) | 1.0 (0.7 to 1.2) | 2.8 (2.0 to 3.5) | 2.5 (1.4 to 4.4) | NA | 12.3 (11.6 to 13.0) | 2.1 (1.9 to 2.4) | 2.9 (2.1 to 3.9) | NA | 16.8 (15.7 to 17.9) |
| Appointments | 6.2 (4.9 to 7.6) | 11.7 (10.3 to 13.2) | 7.3 (5.6 to 9.0) | 10.6 (9.2 to 12.1) | 12.1 (10.9 to 13.3) | 12.0 (9.3 to 15.6) | 8.6 (7.4 to 10.1) | 28.4 (27.4 to 29.4) | 16.6 (16.0 to 17.1) | 14.4 (12.8 to 16.2) | NA | 3.5 (2.9 to 4.1) |
| Any healthcare disruption | 10.1 (8.1 to 12.1) | 15.9 (14.3 to 17.6) | 12.8 (10.3 to 15.4) | 14.3 (12.7 to 15.9) | 16.7 (15.2 to 18.2) | 16.4 (13.2 to 20.2) | 9.4 (8.1 to 10.9) | 31.8 (30.8 to 32.8) | 25.3 (24.6 to 25.9) | 19.9 (18.1 to 21.9) | 6.35 (5.9 to 7.2) | 19.5 (18.3 to 20.8) |

Sources: Millennium Cohort Study (MCS); Children of the Avon Longitudinal Study of Parents and Children (ALSPAC-G1); Next Steps (NS); 1970 British Cohort Study (BCS70); National Child Development Study (NCDS); National Survey of Health and Development (NSHD); Born in Bradford (BIB); Understanding Society (USOC); Generation Scotland: the Scottish Family Health Study (GS); parents of ALSPAC (ALSPAC-G0); UK Adult Twin Registry (TwinsUK); English Longitudinal Study of Ageing (ELSA).
Studies are ordered by age homogeneity/heterogeneity and mean age of respondents at the time of the interview. Samples for each study were restricted to respondents with non-missing information on healthcare disruptions and valid information on sex, social class, education and (where applicable) age and ethnicity. All information about how data were collected and variables were coded is available in online supplemental file 2.
TwinsUK had an additional question: 'Have you experienced healthcare disruption as a result of the COVID-19 pandemic?' These data were also used to derive the 'any healthcare disruption' variable for TwinsUK.
Weighted data where applicable.
NA, not available/info not collected.

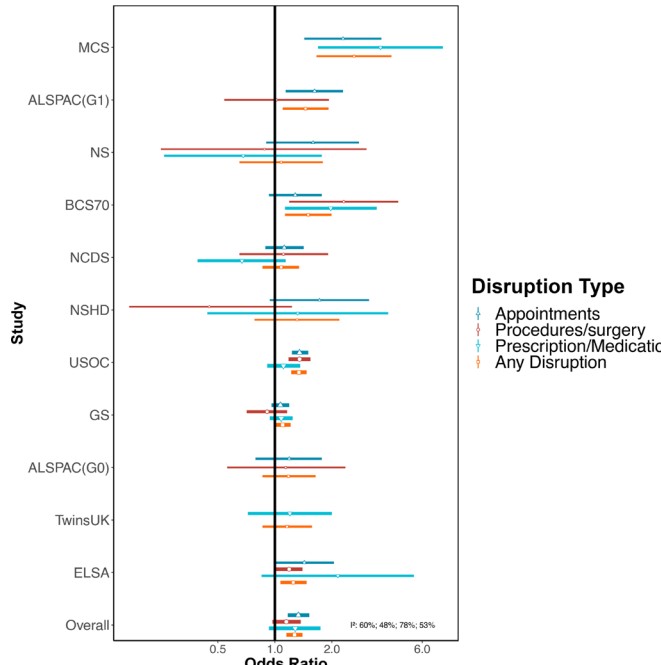

**Figure 2** Associations between female (compared with male) sex and healthcare disruption. Sources: Millennium Cohort Study (MCS); Children of the Avon Longitudinal Study of Parents and Children (ALSPAC-G1); Next Steps (NS); 1970 British Cohort Study (BCS70); National Child Development Study (NCDS); National Survey of Health and Development (NSHD); Understanding Society (USOC); Generation Scotland: the Scottish Family Health Study (GS); parents of ALSPAC (ALSPAC-G0); UK Adult Twin Registry (TwinsUK); English Longitudinal Study of Ageing (ELSA). Adjusted for age and ethnicity where applicable.

medications crossed the null (online supplemental file 3 and figure 2).

There were differences in the association between sex and healthcare disruption when stratified by age (p<0.001, online supplemental file 3). The odds of having any healthcare disruption for females was highest among 16–24 year-olds (OR 2.22; 95% CI 1.63 to 3.02; I²=0%, Supplementary File 3). An association between sex and healthcare disruption was observed up to age 54 years but there were no clear associations among those aged 55 years and above. There was no evidence that the association between sex and healthcare disruption differed by shielding and non-shielding groups (Supplementary File 3).

### Age and healthcare disruptions

A higher prevalence of having any healthcare disruption was observed among older participants of the national birth cohorts where the same questionnaire was used (figure 1). This age difference was also observed among the ALSPAC studies and for other age-heterogenous studies as seen in online supplemental table S4.

The meta-analysis including age-heterogenous studies was supportive of age differences for any healthcare disruptions (eg, OR 1.39; 95% CI 1.13 to 1.72; I²=77% for 65–75 years vs 45–54 years) (figure 3, online supplemental

file 3). Disruptions seemed less likely in younger age groups and more likely among older age groups, though some estimates cross the null and had high heterogeneity, which may be because of few studies in specific age categories (figure 3, online supplemental file 3). Associations for disruptions to medical appointments and procedures or surgery showed these age differences more clearly (figure 3, online supplemental file 3).

There were no clear differences in the association with age and any healthcare disruption by shielding status. However, for those who were shielding, CIs were wide (Supplementary File 3). The magnitude for the association of healthcare disruption among 75 year-olds and above vs 45–54 year-olds was higher among the non-shielding group (OR 1.61; 95% CI 1.17 to 2.22; I²=79%) compared with the shielding group (OR 0.83; 95% CI 0.51 to 1.37; I²=83%, Supplementary File 3).

### Ethnicity and healthcare disruptions

Among the studies that had data on ethnicity, between 7.8% (BIB) and 31.9% (USOC) of the white groups reported healthcare disruption. Between 8.3% (TwinsUK) and 23.6% (GS) of ethnic minority groups reported having any healthcare disruption (online supplemental table S4).

In meta-analysis, ethnic minorities compared with white groups had increased odds of any healthcare disruption (OR 1.19; 95% CI 1.05 to 1.35; I²=0%, figure 4 and Supplementary File 3). This association was less clear for specific domains of healthcare disruption (figure 4, online supplemental file 3). Among the studies that had a finer categorisation of ethnicity, only the black ethnic groups had clearly raised odds for any healthcare disruption compared with white groups (OR 1.38; 95% CI 1.03 to 1.84; I²=0%). Associations with healthcare disruption were less evident for other ethnic groups but were imprecisely estimated (figure 4, online supplemental file 3).

There were no major differences in associations between ethnicity and any healthcare disruption by age, though this may simply be due to low power as CIs were wide (Supplementary File 3). The clearest associations with ethnic minority groups were within the age ranges of 35–44 and 45–74 years (OR 1.31; 95% CI 1.01 to 1.71; I²=0% and OR 1.61; 95% CI 1.16 to 2.22; I²=0%). The mixed ethnicity group was also at particular risk for disruption in the 16–24 years age range (OR 2.50; 95% CI 1.25 to 5.02; I²=0%). The magnitude for the association between any healthcare disruptions among ethnic minority groups versus white groups was higher among those who were shielding (OR 1.56; 95% CI 1.01 to 2.39; compared with OR 1.06; 95% CI 0.86 to 1.31 for non-shielding). This observation was consistent across more granular ethnicity categories, but CIs were wide (Supplementary File 3).

### Education and healthcare disruptions

There was no clear pattern in the prevalence of healthcare disruption across education levels. For example, in

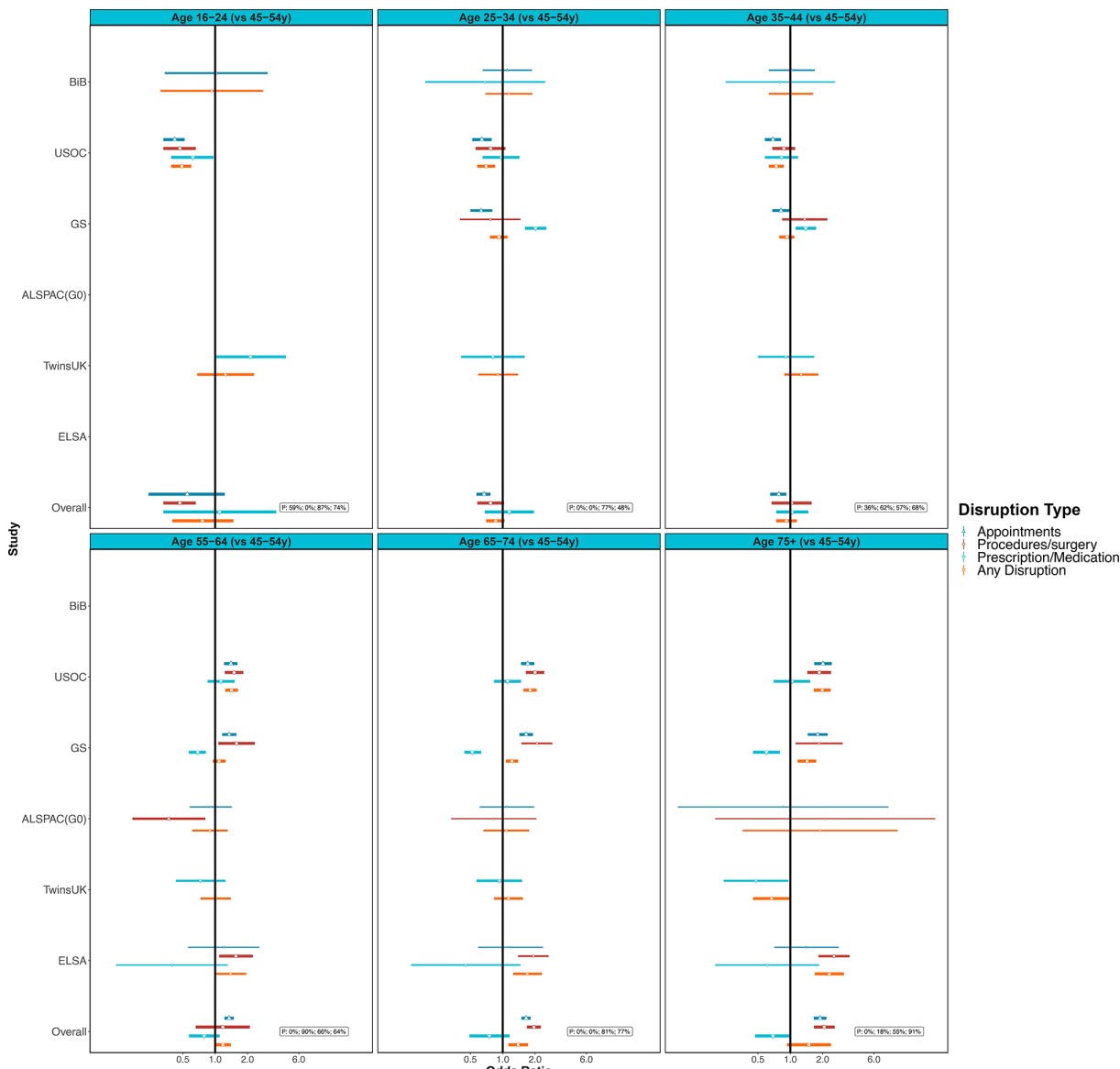

**Figure 3** Associations between age (compared with 45–54 year-olds) and healthcare disruption. Sources: Born in Bradford (BIB); Understanding Society (USOC); Generation Scotland: the Scottish Family Health Study (GS); parents of ALSPAC (ALSPAC-G0); UK Adult Twin Registry (TwinsUK); English Longitudinal Study of Ageing (ELSA). Adjusted for sex and ethnicity where applicable.

USOC 29.7% of those with any healthcare disruption had a degree or equivalent and 39% had no school-leaving qualifications. In TwinsUK, 9.9% of those with any healthcare disruption had a degree or equivalent and 6.1% had no school leaving (online supplemental table S4).

In meta-analysis, we did not observe clear associations between education level and healthcare disruption, other than that those without school-leaving qualifications had raised odds of disruptions to procedures or surgery (OR 1.26; 95% CI 1.11 to 1.44; $I^2$=0%; Supplementary File 3 and figure 5). We did not observe differences by age or shielding status (Supplementary File 3).

### Occupational class and healthcare disruptions

The prevalence of any healthcare disruption ranged from 9.7% (BIB) to 25.7% (USOC) among the professional/ managerial social class and from 9.3% (BIB) to 27.6% (USOC) for the manual/routine social class (online supplemental table S4).

Results from meta-analysis show that those in a more disadvantaged occupational class were more likely to report any healthcare disruptions (eg, OR 1.17; 95% CI 1.08 to 1.27; $I^2$=0% for manual/routine compared with professional/managerial, figure 6, online supplemental file 3). The OR was greatest for the other occupational class category (OR 1.51; 95% CI 1.12 to 2.04); however, the $I^2$ was also large (80%). "The large $I^2$ implies considerable between study heterogeneity. It is worth noting that two of the four individual studies (MCS and ELSA) that did not show clear associations for this category were at the extremes of the age range considered.

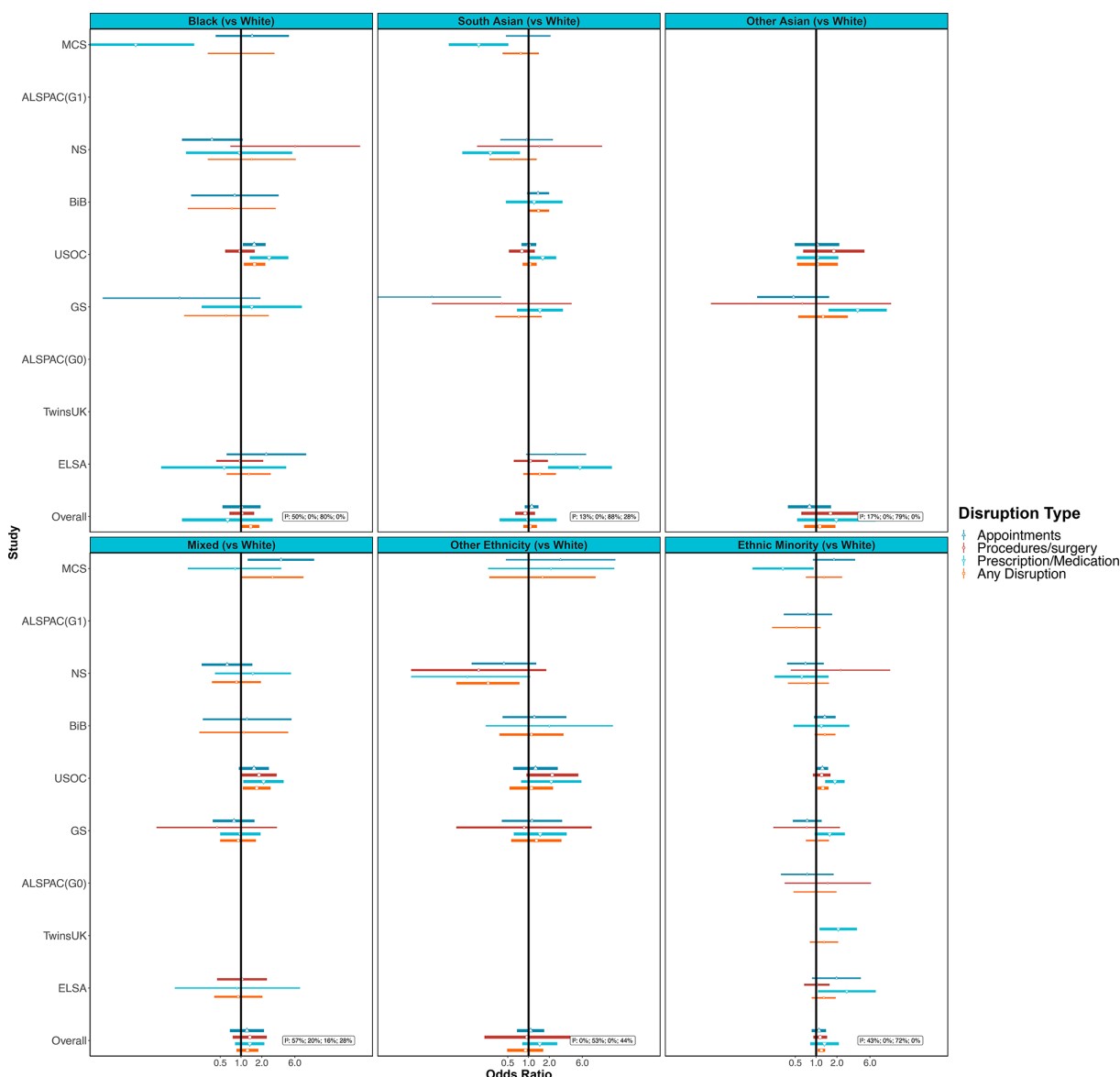

**Figure 4** Associations between ethnicity (compared with white groups) and healthcare disruption. Sources: Millennium Cohort Study (MCS); Children of the Avon Longitudinal Study of Parents and Children (ALSPAC-G1); Next Steps (NS); Born in Bradford (BIB); Understanding Society (USOC); Generation Scotland: the Scottish Family Health Study (GS); parents of ALSPAC (ALSPAC-G0); UK Adult Twin Registry (TwinsUK); English Longitudinal Study of Ageing (ELSA). Panels illustrate findings for some larger ethnic groups separately and the final panel presents results for all non-white ethnic minorities combined. Adjusted for age and sex where applicable.

Similar associations were seen for domains of healthcare disruption, with the largest inequalities seen for access to medications. We did not observe differences by age or shielding status (Supplementary File 3).

### Sensitivity analysis
There were no major differences in the results after restricting to representative samples (Supplementary file 1).

### DISCUSSION
Our study demonstrates marked inequalities in healthcare disruption during the COVID-19 pandemic by harnessing data from 12 UK longitudinal studies. Females were more

likely to report healthcare disruptions than males, especially at younger ages (<55 years). This inequality was observed for each healthcare disruption type including prescription medication, procedures or surgery and appointments as well as a combined measure for any of these disruptions. Older adults were especially likely to report disruptions to medical appointments and procedures and surgeries compared with their younger counterparts. Ethnic minority (excluding white minorities) groups were more likely to report healthcare disruption compared with white (including white minorities) groups. Furthermore, when stratifying results by shielding status, the magnitude for the association between any healthcare disruptions among ethnic minority groups (compared

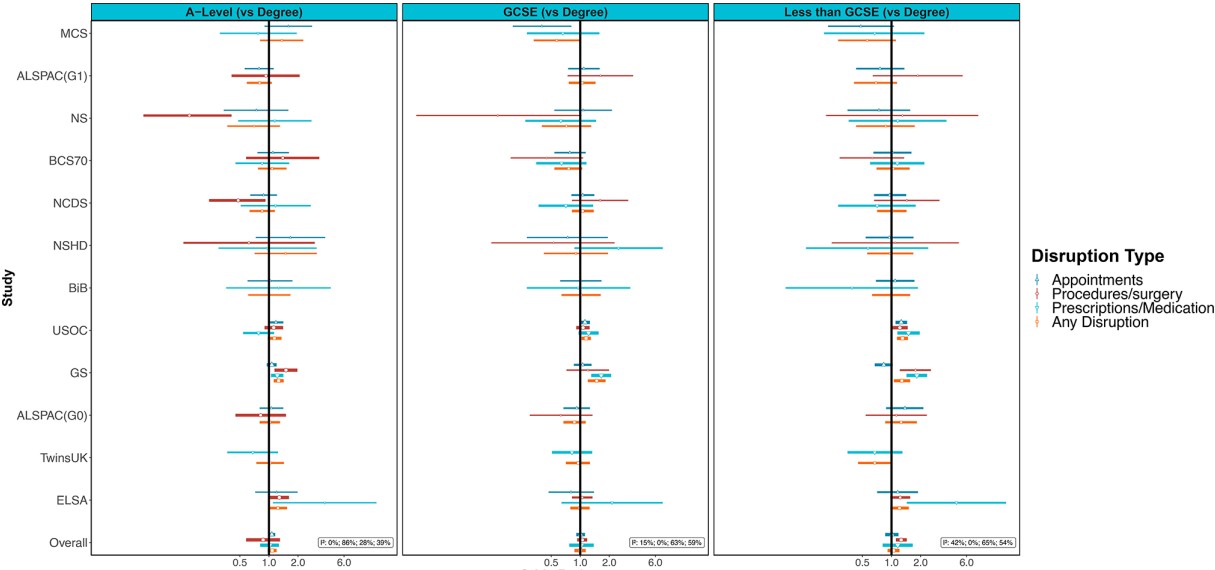

**Figure 5** Associations between education (compared with degree level) and healthcare disruption. GCSE, General Certificate of Secondary Education. Sources: Millennium Cohort Study (MCS); Children of the Avon Longitudinal Study of Parents and Children (ALSPAC-G1); Next Steps (NS); 1970 British Cohort Study (BCS70); National Child Development Study (NCDS); National Survey of Health and Development (NSHD); Born in Bradford (BIB); Understanding Society (USOC); Generation Scotland: the Scottish Family Health Study (GS); parents of ALSPAC (ALSPAC-G0); UK Adult Twin Registry (TwinsUK); English Longitudinal Study of Ageing (ELSA). Adjusted for age, sex and ethnicity where applicable.

with white groups) was higher among those who were shielding. In studies where a finer breakdown of ethnicity was possible, black ethnic minority groups had the most clearly increased odds of disruption compared with white ethnic groups. Occupational class was also found to be associated with healthcare disruption with those in a routine/manual occupation or other (which included never/long-term non-employed) being more likely to

experience healthcare disruption than those in a managerial/professional occupation. No clear association between education and healthcare disruption was found in the main, age or shielding status-stratified analyses.

The direct burden of COVID-19 on health services across the globe has been colossal and remains so in some countries, with prioritisation of patients with COVID-19, leaving less capacity and resources for non-COVID-19

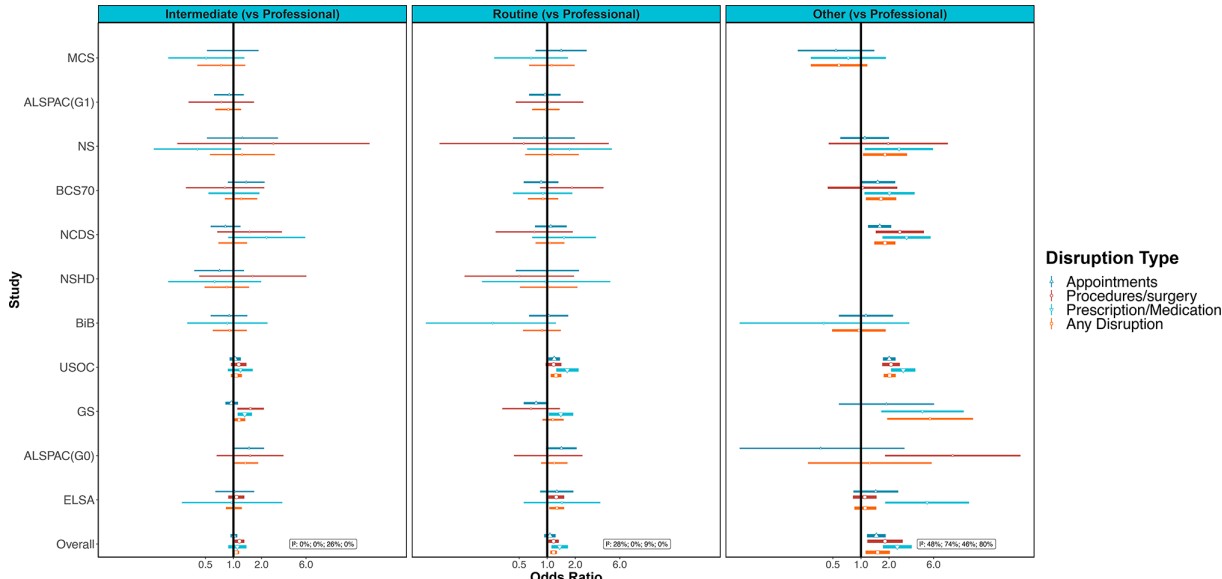

**Figure 6** Associations between occupational social class (compared with professional/managerial) and healthcare disruption. Sources: Millennium Cohort Study (MCS); Children of the Avon Longitudinal Study of Parents and Children (ALSPAC-G1); Next Steps (NS); 1970 British Cohort Study (BCS70); National Child Development Study (NCDS); National Survey of Health and Development (NSHD); Born in Bradford (BIB); Understanding Society (USOC); Generation Scotland: the Scottish Family Health Study (GS); parents of ALSPAC (ALSPAC-G0); English Longitudinal Study of Ageing (ELSA). Adjusted for age, sex and ethnicity where applicable.

healthcare. Furthermore, associated repeated lockdown measures are also likely to decrease healthcare access and availability with a decrease in the number of people attending A&E services,[4] and reports of difficulties accessing medication.[9]

Our findings are consistent with current evidence from a smaller subset of the studies examined here, suggesting that females are more likely to experience disruption to planned surgery, medical procedures or other medical appointments during lockdown.[13] Furthermore, our results show that older adults were more likely to report healthcare disruption as compared with their younger counterparts, especially disruptions to medical appointments and planned procedures or surgeries. This finding is consistent with current UK evidence indicating that older adults experience more delays and disruption to health services.[33–36] Black ethnic minority groups were also found to be at increased risk of healthcare disruption compared with white ethnic groups—an issue of particular concern given prepandemic ethnic inequalities in healthcare.[37] The inequalities by occupational class we found are consistent with prior evidence of socioeconomic healthcare inequalities reported in the UK in the past decade,[38] and highlight that these have still been present in the COVID-19 pandemic. Associations with occupational class were clearer than those for education, which is also an indicator of socioeconomic position but may have been a more distal influence.

The sex inequalities observed in this study could partially be explained by a disproportional increase in childcare responsibilities for women,[39] which may have made it more difficult to access healthcare. However, in this study we adjusted for household composition and associations for sex were robust to further adjustment on this variable.

Our results also show that older adults were more likely to report healthcare disruption than younger adults. There are many reasons why older people may have experienced an increase in healthcare disruption during the COVID-19 pandemic compared with younger people, including fear of becoming infected while visiting a care facility, difficulties engaging in telemedicine (using technology to deliver care)[33–35] and greater frailty, resulting in more healthcare utilisation and subsequent disruption.[36]

One explanation for the inequality in healthcare disruption among black ethnic minority group may be due to adverse effects of loss of income, unstable housing, increased psychological distress and reduced community support brought about by lockdown restrictions. Another explanation could stem from a disproportionate representation of ethnic minority populations among key workers, who are subjected to increased and antisocial working hours.

## Strengths and limitations

The analysis brings together data from 12 longitudinal studies with rich and sensitive information on healthcare disruption. This study is strengthened by the coordinated investigation in multiple longitudinal studies with differing study designs, different target populations and varying selection and attrition processes. Our combined approach provides the largest sample size available to prospectively investigate differences between ethnic groups, within representative population-based samples. What's more, though using non-response weights available, the proportion of ethnic minority groups within most studies is representative of the UK population. Moreover, the use of multiple studies increased statistical power to look at subpopulations such as ethnic minority groups across cohorts and allowed for greater examination of how inequalities were patterned by age. While not all 12 studies were representative of the population of interest, removing them in sensitivity analyses did not change our conclusions. Our novel approach to coordinated analyses harnessing multiple data sets therefore allowed research questions to be addressed which would not otherwise be possible.

Differences between studies in a range of factors including measurement of healthcare disruption, timing of surveys, design, response rates and differential selection into the COVID-19 sweeps are potentially responsible for heterogeneity in estimates. However, despite this heterogeneity, the key findings were consistent across most data sets. Furthermore, this heterogeneity can be informative, for example, by virtue of mixing age-specific and age range studies, we identified that sex inequalities were stronger at younger ages. The definition of healthcare disruption used may also have contained a range of disruptions of greater or lesser severity, and there may have been further inequalities in the severity of disruptions experienced; however, we were not able to assess this using the available data. We also could not assess prepandemic inequalities in healthcare disruption, though other studies have indicated massive increases in the prevalence of healthcare disruption (at least in part from the supply side with non-urgent procedures cancelled to reduce risk of infection transmission), and that inequalities related to geographic measures of deprivation (rather than individual-level measures as used here) have widened during the pandemic.[5 40 41]

We have focused on our aim of identifying who experienced greater disruptions in healthcare, rather than on adjustment for confounders to estimate causal effects of the exposures in question.[42] Nevertheless, many of the associations we observed were robust to adjustment for a wider range of related variables, but bias due to residual confounding cannot be ruled out. Importantly, we did not condition our analyses on healthcare need. Many of the inequalities we observed for healthcare disruptions may be due to inequalities in health, with those who have greater health needs being more likely to require healthcare that could be disrupted. Accounting for differences in need could have masked inequalities in healthcare disruptions that are caused by inequalities in health and could have made it less clear which groups have been more likely to experience disruption during the

pandemic. Restricting analyses to those who needed care could also induce bias if there were unmeasured determinants of both need and disruption.[43] Nevertheless, another study of the USOC data analysed here that did restrict analyses to those needing care still found income-related inequalities in healthcare disruption, and most of the associations we observed were robust to adjustment for prepandemic self-assessed health.[44]

### Impact of healthcare disruption

Disadvantaged groups such as females, older adults, black ethnic minority groups and those in routine/manual occupations have had elevated odds of healthcare disruption in the first 8–10 months of the COVID-19 pandemic.

Delays and disruptions to treatment could have ongoing implications for patients' physical and mental health.[45] Action is needed to remedy these inequalities, and efforts to ensure continuity of care during pandemic-related disruptions may need to be more clearly targeted to those who most need that care. Actions to alleviate healthcare disruption inequalities critically rely on better understanding the causes. For example, barriers to accessing care, such as working hours or fear of infection, may require measures to make care more accessible outside of working hours, or to increase public confidence that patients can attend safely.

As healthcare access resumes, given the forgone delays in treatments and the subsequent backlog of postponed surgeries,[46] these groups may require prioritised support to address unmet needs experienced during the pandemic.

### CONCLUSION

There have been clear inequalities in disruptions to healthcare during the COVID-19 pandemic in the UK. Females (especially those aged 54 or younger), older adults, ethnic minorities and those in disadvantaged occupational classes have been more likely to experience healthcare disruptions. These are groups who usually experience worse health, so considering the massive increases in the prevalence of healthcare disruptions related to COVID-19, these inequalities in disruption have clear potential to maintain or even exacerbate existing health inequalities.

**Author affiliations**
¹MRC Unit for Lifelong Health and Ageing, UCL, London, UK
²Centre for Longitudinal Studies, Social Research Institute, UCL, London, UK
³Department of Epidemiology and Public Health, UCL, London, UK
⁴MRC/CSO Social & Public Health Sciences Unit, University of Glasgow, Glasgow, UK
⁵Department of Twin Research and Genetic Epidemiology, School of Life Course & Population Sciences, King's College London, London, UK
⁶Centre for Genomic and Experimental Medicine, Institute of Genetics and Cancer, University of Edinburgh, Edinburgh, UK
⁷Division of Psychiatry, University of Edinburgh, Edinburgh, UK
⁸MRC Integrative Epidemiology Unit, University of Bristol, Bristol, UK
⁹Department of Neuroscience, Psychology and Behaviour, University of Leicester, Leicester, UK
¹⁰Bradford Teaching Hospitals NHS Foundation Trust, Bradford, UK

**Acknowledgements** The contributing studies have been made possible because of the tireless dedication, commitment and enthusiasm of the many people who have taken part. We would like to thank the participants and the numerous team members involved in the studies including interviewers, technicians, researchers, administrators, managers, health professionals and volunteers, including: Generation Scotland: Drew Altschul, Chloe Fawns-Ritchie, Archie Campbell, Robin Flaig; ALSPAC: Daniel J Smith, Nicholas J Timpson, Kate Northstone; Understanding Society: Michaela Benzeval; TwinsUK: Deborah Hart, María Paz García, Rachel Horsfall, Ruth C E Bowyer; Centre for Longitudinal Studies: Matt Brown, Lisa Calderwood, Emla Fitzsimons, Alissa Goodman, Aida Sanchez; NSHD: Andrew Wong, Maria Popham, Karen MacKinnon, Imran Shah, Philip Curran. We are extremely grateful to all the families who took part in this study, together with the interviewers, computer and laboratory technicians, clerical workers, research scientists, volunteers, managers, receptionists and nurses. We are additionally grateful to our funders for their financial input and support in making this research happen.

**Contributors** SVK, GBP, JM, SP, GDG, MJG and EJT conceptualised the study and design. SVK, GBP, JM, SP, GDG, MJG, EJT and RJS designed the methodology. JM, SP, GDG, MJG, EJT, AJS, ASFK, EM and GS conducted the formal analysis. JM, SP, GDG, MJG and EJT drafted the manuscript. All authors contributed to critical revision and provided final approval of the manuscript. The project was supervised by GBP and SVK. Funding was acquired by PP, SVK, GBP, RJS and NC. JM, SP, GSG, MJG, EJT, ASFK, EM, GS, GBP and SVK are responsive for the overall content and guarantor.

**Funding** This work was supported by the National Core Studies, an initiative funded by UKRI, NIHR and the Health and Safety Executive. The COVID-19 Longitudinal Health and Well-being National Core Study was funded by the Medical Research Council (MC_PC_20030). Understanding Society is an initiative funded by the Economic and Social Research Council and various government departments, with scientific leadership by the Institute for Social and Economic Research, University of Essex, and survey delivery by NatCen Social Research and Kantar Public. The Understanding Society COVID-19 Study is funded by the Economic and Social Research Council (ES/K005146/1) and the Health Foundation (2076161). The research data are distributed by the UK Data Service. The Millennium Cohort Study, Next Steps, the 1970 British Cohort Study and 1958 National Child Development Study are supported by the Centre for Longitudinal Studies, Resource Centre 2015-20 grant (ES/M001660/1) and a host of other cofunders. The 1946 NSHD cohort is hosted by the MRC Unit for Lifelong Health and Ageing funded by the Medical Research Council (MC_UU_00019/1 Theme 1: Cohorts and Data Collection). The COVID-19 data collections in these five cohorts were funded by the UKRI grant: Understanding the economic, social and health impacts of COVID-19 using lifetime data: evidence from 5 nationally representative UK cohorts (ES/V012789/1). The English Longitudinal Study of Ageing was developed by a team of researchers based at University College London, NatCen Social Research, the Institute for Fiscal Studies, the University of Manchester and the University of East Anglia. The data were collected by NatCen Social Research. The funding is currently provided by the National Institute on Aging (Ref: R01AG017644) and by a consortium of UK government departments: Department for Health and Social Care; Department for Transport; Department for Work and Pensions, which is coordinated by the National Institute for Health Research (NIHR, Ref: 198-1074). Funding has also been provided by the Economic and Social Research Council (ESRC).The UK Medical Research Council and Wellcome Trust (grant reference: 217065/Z/19/Z) and the University of Bristol provide core support for ALSPAC. A comprehensive list of grants funding is available on the ALSPAC website (http://www.bristol.ac. uk/alspac/external/documents/grant-acknowledgements.pdf). TwinsUK receives funding from the Wellcome Trust (WT212904/Z/18/Z), the National Institute for Health Research (NIHR) Biomedical Research Centre based at Guy's and St Thomas' NHS Foundation Trust and King's College London. TwinsUK is also supported by the Chronic Disease Research Foundation and Zoe Global. Generation Scotland received core support from the Chief Scientist Office of the Scottish Government Health Directorates (CZD/16/6) and the Scottish Funding Council (HR03006). Genotyping of the GS:SFHS samples was carried out by the Genetics Core Laboratory at the Wellcome Trust Clinical Research Facility, Edinburgh, Scotland, and was funded by the Medical Research Council UK and the Wellcome Trust (Wellcome Trust Strategic Award 'STratifying Resilience and Depression Longitudinally' (STRADL); reference: 104036/Z/14/Z). Generation Scotland is funded by the Wellcome Trust (216767/Z/19/Z). Born in Bradford (BIB) receives core infrastructure funding from the Wellcome Trust (WT101597MA), and a joint grant from the UK Medical Research Council (MRC) and UK Economic and Social Science Research Council (ESRC) (MR/N024397/1), and one from the British Heart Foundation (BHF) (CS/16/4/32482). The National Institute for Health Research Yorkshire and Humber

ARC and the Clinical Research Network both provide support for BiB research. SVK acknowledges funding from an NRS Senior Clinical Fellowship (SCAF/15/02), the Medical Research Council (MC_UU_00022/2) and the Scottish Government Chief Scientist Office (SPHSU17). ASFK acknowledges funding from the ESRC (ES/V011650/1). EJT acknowledges funding from Wellcome Trust (WT212904/Z/18/Z). GBP acknowledges funding from the Economic and Social Research Council (ES/V012789/1). GC acknowledges funding from British Heart Foundation Special Programme Grant SP/20/2/34841

**Disclaimer** The funders had no role in study design, data collection and analysis, decision to publish or preparation of the manuscript.

**Competing interests** SVK is a member of the Scientific Advisory Group on Emergencies subgroup on ethnicity and COVID-19 and is cochair of the Scottish Government's Ethnicity Reference Group on COVID-19. NC serves on a data safety monitoring board for trials sponsored by AstraZeneca.

**Patient and public involvement** Patients and/or the public were not involved in the design, or conduct, or reporting, or dissemination plans of this research.

**Patient consent for publication** Not applicable.

**Ethics approval** Ethics statement and data access details for each study can be found in online supplemental table S2 in supplementary file 4.

**Provenance and peer review** Not commissioned; externally peer reviewed.

**Data availability statement** Data are available upon reasonable request. Data for NCDS (SN 6137), BCS70 (SN 8547), Next Steps (SN 5545), MCS (SN 8682) and all four COVID-19 surveys (SN 8658) are available through the UK Data Service. NSHD data are available on request to the NSHD Data Sharing Committee. Interested researchers can apply to access the NSHD data via a standard application procedure. Data requests should be submitted to mrclha.swiftinfo@ucl.ac.uk; further details can be found at http://www.nshd.mrc.ac.uk/data.aspx. doi:10.5522/NSHD/Q101; doi:10.5522/NSHD/Q10. The ALSPAC study website contains details of all the data that is available through a fully searchable data dictionary and variable search tool: http://www.bristol.ac.uk/alspac/researchers/our-data. ALSPAC data is available to researchers through an online proposal system. Information regarding access can be found on the ALSPAC website (http://www.bristol.ac.uk/media-library/sites/alspac/documents/researchers/data-access/ALSPAC_Access_Policy.pdf). Data from the various BiB family studies are available to researchers; see the study website for information on how to access data (https://borninbradford.nhs.uk/research/how-to-access-data/). All USOC data are available through the UK Data Service (SN 6614 and SN 8644). All ELSA data are available through the UK Data Service (SN 8688 and 5050). Access to data from GS is approved by the Generation Scotland Access Committee. See https://www.ed.ac.uk/generation-scotland/for-researchers/access or email access@generationscotland.org for further details. The TwinsUK Resource Executive Committee (TREC) oversees management, data sharing and collaborations involving the TwinsUK registry (for further details see https://twinsuk.ac.uk/resources-for-researchers/access-our-data/).

**ORCID iDs**
Jane Maddock http://orcid.org/0000-0002-7975-4221
Sam Parsons http://orcid.org/0000-0001-5949-3899
Giorgio Di Gessa http://orcid.org/0000-0001-6154-1845
Michael J Green http://orcid.org/0000-0003-3193-2452
Ellen J Thompson http://orcid.org/0000-0003-2118-821X
Anna J Stevenson http://orcid.org/0000-0002-0435-3562
Alex SF Kwong http://orcid.org/0000-0003-1953-2771
Gillian Santorelli http://orcid.org/0000-0003-0427-1783
Richard J Silverwood http://orcid.org/0000-0002-2744-1194
Gabriella Captur http://orcid.org/0000-0002-5662-0642
Nishi Chaturvedi http://orcid.org/0000-0002-6211-2775
Claire J Steves http://orcid.org/0000-0002-4910-0489
Andrew Steptoe http://orcid.org/0000-0001-7808-4943
Praveetha Patalay http://orcid.org/0000-0002-5341-3461
George B Ploubidis http://orcid.org/0000-0002-8198-5790
Srinivasa Vittal Katikireddi http://orcid.org/0000-0001-6593-9092

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
