## [Reviewer comments · BMJ Open]

ARTICLE DETAILS

TITLE (PROVISIONAL)	Inequalities in healthcare disruptions during the COVID-19 pandemic: evidence from 12 UK population-based longitudinal studies
AUTHORS	Maddock, Jane; Parsons, Sam; Di Gessa, Giorgio; Green, Michael; Thompson, Ellen J.; Stevenson, Anna J.; Kwong, Alex; McElroy, Eoin; Santorelli, Gillian; Silverwood, Richard; Captur, Gaby; Chaturvedi, Nishi; Steves, Claire J.; Steptoe, Andrew; Patalay, Praveetha; Ploubidis, George; Katikireddi, Srinivasa

VERSION 1 – REVIEW

REVIEWER	Kevin Callison
REVIEW RETURNED	20-Jun-2022

GENERAL COMMENTS	This paper examines associations between individual demographic characteristics (e.g., age, sex, race and ethnicity) and care disruptions due to COVID-19. A particular strength of this study is that it aggregates responses from 12 different longitudinal surveys allowing for improved statistical power and the ability to examine the consistency of the results across different surveys. The authors find that females, racial and ethnic minorities, and older people were more likely to report health care disruptions during the pandemic. However, as noted in the limitations discussion, the greater degree of care disruption for these demographic groups may, in part, reflect their higher baseline demand for care. Overall, this study contributes to the emerging literature on COVID-19 and disruptions to care. I have only a few minor comments that I'll list below: 1. The last sentence of the Design section reads, "we can provide robust evidence to understand how the pandemic has impacted population health and support efforts to mitigate its health effects going forward." This statement implies causality ("impacted") and also implies that the authors' study health outcomes in addition to access/care disruption outcomes (which they do not do).2. Additional clarity on identifying those who were advised to "shield" would be helpful. Did the surveys directly ask respondents whether they were advised to shield or did the authors identify this population based on other survey questions?3. I would also appreciate some additional clarification around survey timing and how results are aggregated across different time horizons. According to Supplementary Table S1, the 12 surveys used in the study were fielded at various times throughout 2020. I would expect the magnitude of care disruptions (and perhaps their association with demographic characteristics) to be different at different points in time (i.e., number of months from the beginning of the pandemic or from various lockdown periods). How do the authors deal with this issue? More information would be helpful.4. Most of the Discussion section is devoted to summarizing the
---

	findings, with only a short paragraph discussing implications. I think the authors could spend more time on interpreting their findings and providing context for any policy implications going forward.
--	--

REVIEWER	Alberto Mateo-Urdiales Istituto Superiore di Sanità, Infectious Diseases Department
REVIEW RETURNED	19-Jul-2022

GENERAL COMMENTS	Thank you for this review on a very important topic. I found the paper interesting and easy to read. I have a couple of major comments and some minor ones. Major comments  1. You report the response rates for each survey (Sup T1), but you do not report the response rate for the questions assessed in your study. Given that you only included non-missing data in the analysis. The reader might find informative how many people actually responded to the question. 2. The main issue I have is with the conclusions reported in the main text and the abstract, which I am not sure they follow from the results. You assessed inequalities in disruption to healthcare during the pandemic, but without assessing the pre-existing gap you cannot conclude that they "maintain or widen inequalities". This would be the conclusion only if the same data tells you that before the pandemic the inequalities were narrower. Is this the case? Why did you not assess changes in inequalities pre- and during the pandemic? I guess the answer to the latter question is that it was not possible given that the questions in the survey relate to the pandemic, but this does not look always the case. Let's take questions regarding "prescription or medication access" (Sup File 1). Only two questions ask explicitly if the disruptions are caused or made worse by the pandemic (GS and TWINS). Others, it may be argued, do it implicitly as they are asking about COVID (e.g. BIB, ELSA). But, for example USOC seems to ask the question without mentioning the pandemic. In fact, USOC seems to ask respondents to often compare their situation now with the previous survey. Was this the only study that enabled the assessment of inequalities through time? (I know that you do mention the USOC study (ref 41) in the limitations. However, the study relates to income inequalities, which was not an indicator in your study.) So, I would suggest not to conclude that inequalities are the same or worse than before the pandemic, as you cannot infer that with your results. Minor comments  1. In the abstract you report an I2 of 53% for the female vs male analysis, but in the table of SF2, the figure is 54%. 2. You mention in the methods (page 6, line 150) that "where respondents' education and occupational class were not available, we considered parental education or household social class." In how many did this happen? Could you provide these figures in the text? 3. In the conclusion (page 20, line 404) you mention that "Females (especially at younger ages) Were more likely to experience healthcare disruption". However, according to your result it was only females aged 54 or less. Thus, it is not "especially younger ones"
--

(though there is a gradient), but “only those aged 54 or less”.

VERSION 1 – AUTHOR RESPONSE

Reviewer: 1
Kevin Callison

This paper examines associations between individual demographic characteristics (e.g., age, sex, race and ethnicity) and care disruptions due to COVID-19. A particular strength of this study is that it aggregates responses from 12 different longitudinal surveys allowing for improved statistical power and the ability to examine the consistency of the results across different surveys. The authors find that females, racial and ethnic minorities, and older people were more likely to report health care disruptions during the pandemic. However, as noted in the limitation's discussion, the greater degree of care disruption for these demographic groups may, in part, reflect their higher baseline demand for care. Overall, this study contributes to the emerging literature on COVID-19 and disruptions to care. I have only a few minor comments that I'll list below:

Response: Thank you for acknowledging the strengths of our study.

1. The last sentence of the Design section reads, "we can provide robust evidence to understand how the pandemic has impacted population health and support efforts to mitigate its health effects going forward." This statement implies causality ("impacted") and also implies that the authors' study health outcomes in addition to access/care disruption outcomes (which they do not do).

*Response: Thank you. We have edited this sentence as follows:
"By coordinating analyses within each study and statistically pooling results in a meta-analysis, we can provide robust evidence to understand healthcare disruptions during the pandemic."*

2. Additional clarity on identifying those who were advised to "shield" would be helpful. Did the surveys directly ask respondents whether they were advised to shield or did the authors identify this population based on other survey questions?

*Response: We have clarified the wording for this measure as follows:
"Respondents were directly asked whether they had received a letter from the NHS advising them to stay at home and protect themselves. Specific survey questions can be found in Supplementary File 1."*

3. I would also appreciate some additional clarification around survey timing and how results are aggregated across different time horizons. According to Supplementary Table S1, the 12 surveys used in the study were fielded at various times throughout 2020. I would expect the magnitude of care disruptions (and perhaps their association with demographic characteristics) to be different at different points in time (i.e., number of months from the beginning of the pandemic or from various lockdown periods). How do the authors deal with this issue? More information would be helpful.

*Response: Surveys were fielded at various times, for example, we stated:
"Where multiple pandemic survey waves had been included, we coded for any disruptions reported up to and including the most recent. This meant at least 7 months of follow-up for most studies (GS had five and ELSA four, while ALSPAC had the longest follow-up period at nine months)."
We deal with this heterogeneity in study methodology, in much the same way as we deal with other variations in methodology (such as question wording etc.), that is, by pooling the results using a random-effects meta-analysis. The resulting meta-analysis estimates could be interpreted as results that are averaged across these variations in study methodology, and despite some heterogeneity in the meta-analyses, the main study estimates were largely consistent with the reported findings. In order to provide more information, we have added a supplementary table showing how the prevalence of disruptions reported during the pandemic changes over time across the six waves of the Understanding Society survey, and include the following text in the methods:
"Supplementary Table S3 shows how the prevalence for any experience of each disruption accumulated across the six USOC surveys. The majority of those who experienced each type of healthcare disruption had already experienced it by the end of May 2020."*

4. Most of the Discussion section is devoted to summarizing the findings, with only a short paragraph discussing implications. I think the authors could spend more time on interpreting their findings and providing context for any policy implications going forward.

Response: Thank you. We have now extended this section and interpreted the findings in more detail:

“Disadvantaged groups such as females, older adults, Black ethnic minority groups, and those in routine/manual occupations have had elevated odds of healthcare disruption in the first 8-10 months of the COVID-19 pandemic.³⁹ Delays and disruptions to treatment could have ongoing implications for patients’ physical and mental health.⁴⁸ Action is needed to remedy these inequalities, and efforts to ensure continuity of care during pandemic-related disruptions may need to be more clearly targeted to those who most need that care. Actions to alleviate healthcare disruption inequalities critically rely on better understanding the causes. For example, barriers to accessing care such as working hours or fear of infection, may require measures to make care more accessible outside of working hours, or to increase public confidence that patients can attend safely.

As healthcare access resumes, given the forgone delays in treatments and the subsequent backlog of postponed surgeries,⁴⁹ these groups may require prioritised support to address unmet needs experienced during the pandemic.”

Reviewer: 2

Dr. Alberto Mateo-Urdiales, Istituto Superiore di Sanità

Comments to the Author:

Thank you for this review on a very important topic. I found the paper interesting and easy to read. I have a couple of major comments and some minor ones.

Major comments

1. You report the response rates for each survey (Sup T1), but you do not report the response rate for the questions assessed in your study. Given that you only included non-missing data in the analysis. The reader might find informative how many people actually responded to the question.

Response: Thank you. We agree that some summary information regarding this issue is important and have included the following statement:

“Missing data within surveys was generally low, especially for healthcare disruption variables, but approximately 5-10% of respondents across studies were excluded due to missing baseline covariates.”

2. The main issue I have is with the conclusions reported in the main text and the abstract, which I am not sure they follow from the results. You assessed inequalities in disruption to healthcare during the pandemic, but without assessing the pre-existing gap you cannot conclude that they "maintain or widen inequalities". This would be the conclusion only if the same data tells you that before the pandemic the inequalities were narrower. Is this the case? Why did you not assess changes in inequalities pre- and during the pandemic?

I guess the answer to the latter question is that it was not possible given that the questions in the survey relate to the pandemic, but this does not look always the case. Let's take questions regarding "prescription or medication access" (Sup File 1). Only two questions ask explicitly if the disruptions are caused or made worse by the pandemic (GS and TWINS). Others, it may be argued, do it implicitly as they are asking about COVID (e.g. BIB, ELSA). But, for example USOC seems to ask the question without mentioning the pandemic. In fact, USOC seems to ask respondents to often compare their situation now with the previous survey. Was this the only study that enabled the assessment of inequalities through time? (I know that you do mention the USOC study (ref 41) in the limitations. However, the study relates to income inequalities, which was not an indicator in your study.)

So, I would suggest not to conclude that inequalities are the same or worse than before the pandemic, as you cannot infer that with your results.

Response: First of all, we should clarify that pre-pandemic data on healthcare disruption were not available across all surveys, and the USOC questions were asking about disruptions since the previous COVID survey (with the April 2021 survey questions covering back to the start of the pandemic). More importantly, though, it was not our intention to assert the inequalities in healthcare disruption had widened. Our point was actually about inequalities in health, which, given the scale of healthcare disruptions during the pandemic and the inequalities in healthcare disruption that we observed during the pandemic, are likely to be maintained or widened.

In order to express this more clearly, we have added the following to the limitations:

“We also could not assess pre-pandemic inequalities in healthcare disruption, though other studies have indicated massive increases in the prevalence of healthcare disruption (at least in part from the supply side, with non-urgent procedures cancelled to reduce risk of infection transmission), and that inequalities related to geographic measures of deprivation (rather than individual-level measures as used here) have widened during the pandemic.”

And re-worded the Conclusions as follows:

“These are groups who usually experience worse health, so considering the massive increases in the prevalence of healthcare disruptions related to COVID-19, these inequalities in disruption have clear potential to maintain or even exacerbate existing health inequalities.”

Minor comments

1. In the abstract you report an I2 of 53% for the female vs male analysis, but in the table of SF2, the figure is 54%.

Response: Thank you for spotting this error. Edited to 54%.

2. You mention in the methods (page 6, line 150) that “where respondents’ education and occupational class were not available, we considered parental education or household social class.” In how many did this happen? Could you provide these figures in the text?

Response: Edited to read as follows:

“Respondents’ education and occupational class were not available in the MCS or ALSPAC-G1 due to the younger age of these cohorts, so we considered parental education or household social class.”

3. In the conclusion (page 20, line 404) you mention that “Females (especially at younger ages) ... Were more likely to experience healthcare disruption”. However, according to your result it was only females aged 54 or less. Thus, it is not “especially younger ones” (though there is a gradient), but “only those aged 54 or less”.

VERSION 2 – REVIEW

REVIEWER	Kevin Callison
REVIEW RETURNED	16-Sep-2022

GENERAL COMMENTS	The authors have adequately addressed the comments from my prior review and I recommend publication of the study.
---

REVIEWER	Alberto Mateo-Urdiales Istituto Superiore di Sanità, Infectious Diseases Department
REVIEW RETURNED	14-Sep-2022

GENERAL COMMENTS	Thanks. All points seem addressed
-----------------------------------